# Functioning of Children and Adolescents with Cancer

**DOI:** 10.3390/ijerph19159762

**Published:** 2022-08-08

**Authors:** Olívia Lopes, Jaqueline Frônio, Anke Bergmann, Rayla Lemos, Érica Defilipo, Paula Chagas

**Affiliations:** 1Graduate Program in Rehabilitation Sciences and Physical Functional Performance, Physical Therapy School, Universidade Federal de Juiz de Fora (UFJF), Juiz de Fora 36038-330, MG, Brazil; 2Molecular Carcinogenesis Program, Instituto Nacional de Câncer (INCA), Rio de Janeiro 20231-050, RJ, Brazil; 3Physical Therapy School, Universidade Federal de Juiz de Fora (UFJF), Juiz de Fora 36038-330, MG, Brazil; 4Physical Therapy Department, Universidade Federal de Juiz de Fora (UFJF)—Governador Valadares Campus, Governador Valadares 35012-000, MG, Brazil

**Keywords:** cancer, childhood, functioning, mobility, physical therapy

## Abstract

The aim of the current study was to evaluate the functioning of children and adolescents diagnosed with cancer. This was a cross-sectional, observational study, with children and adolescents diagnosed with cancer, from 2 to 18 years of age, of both sexes, invited to participate in the city of Juiz de Fora, Minas Gerais, Brazil. The Pediatric Evaluation of Disability Inventory-Computer Adaptive Test (PEDI-CAT) questionnaire was applied to the caregivers to assess the functioning of the participants, in four domains: daily activities, mobility, social/cognitive, and responsibility. In total, 33 children and adolescents participated, of both sexes, with a mean age of 9.23 years. The results showed that in the mobility category, participants older than 8 years presented worse functioning (OR = 22.000, 95% CI = 3.415–141.733, *p* = 0.0001). Children older than 8 years of age and adolescents with different types of cancer showed a higher chance of presenting lower mobility than their normal peers of the same age and compared with children under 8 years of age. Understanding the impact of childhood cancer is important for the physiotherapist to determine treatment strategies for this population who live with dysfunctions left by the cancer treatment.

## 1. Introduction

Childhood cancer is the uncontrolled proliferation of abnormal cells and can occur anywhere in the body [1]. The most common cancers in childhood and adolescence are leukemias, followed by those that affect the central nervous system and lymphomas. In Brazil and in developing countries, cancer is the leading cause of death due to illness among children and adolescents aged 1 to 19 years of age [2].

The treatment of childhood cancer is complex and includes several stages and forms of treatment, in isolation or combined, such as surgery, radiotherapy, and chemotherapy, among others [3]. Of these, chemotherapy is the most common and is a set of drugs that acts at various stages of cellular metabolism, reaching beyond the malignant cells, to the healthy cells in the organism, and being responsible for several reactions [4,5]. All these factors and reactions cause discomfort, suffering, and stress, as well as prolonged hospital stays. Therefore, the child cannot participate actively in the main areas of life, including the home, school, and community [3].

From the moment of diagnosis, during the treatment and its complications, the cancer patient and their relatives experience highly stressful situations, in both physical and emotional domains. The impact of this new reality can generate anxiety, depression, irritability, disorientation, loss of control, and fear of death [6,7]. The child is required to deal with uncertainty about the future and with the feeling of loss of control. This leads to dependency of care from others for many tasks which used to be performed alone, precipitating loss of autonomy in self-care. The change in the child’s routine caused by the new treatment leads to limitations in activities and overprotection from parents [7,8,9,10].

The way the child deals with cancer varies according to their age, contextual factors (i.e., personal, and environmental factors), impairments in body structures and functions, limitations in activity, and restrictions in participation [11,12]. The protective attitude of the family, while often representing a positive factor, may also present a barrier to the child’s interaction with the environment, including school. This health condition is a stressful factor for the whole family, modifying their attitudes. Depending on the evolution of the disease, the impact of all these factors can change the development of the functioning of the child [13,14].

It is still poorly understood how the functioning of children and adolescents is affected by cancer. In addition, there is a lack of studies to guide the physiotherapeutic procedures essential for the maintenance, development, preservation, and improvement in functional capacity of cancer patients. Therefore, the objective of the current study was to evaluate the functioning of children and adolescents diagnosed with cancer and verify the association of the following factors: socioeconomic and demographic variables, type of cancer, type and stages of treatment, physical therapy, and type of health care with functioning.

## 2. Materials and Methods

### 2.1. Participants

All children and adolescents between the ages of 2 and 18 years with a diagnosis of cancer were invited to participate. They could be or not in the medical treatment phase between August 2018 and May 2019. The local children and adolescent cancer treatment institution was contacted, which provided a list with the demographic data of all assisted patients. This study was approved by the Committee on Ethics in Research with Human Beings of the Federal University of Juiz de Fora (CAAE: 82561518.6.0000.5147) and all participants and their guardians signed the informed consent form. Children and adolescents with a non-cancer-related neurological syndrome, with a diagnosis of depression, who refused to answer the questionnaires, and patients who did not know they had cancer, were excluded.

### 2.2. Study Design

An observational cross-sectional study was conducted in the city of Juiz de Fora, Minas Gerais, Brazil.

### 2.3. Setting and Control Variables

Telephone contact was made with all the possible participants and an interview date was scheduled at the most convenient place for the family. Participants and their caregivers who agreed to participate in the study filled out a personal and environmental factsheet including the following control variables, stratified for analysis in the following categories:

Sex: female or male;Age: below 8 years, or above 8 years-The age of 8 years was used as a cut-off age to stratify the participants because this age was close to the mean age of the participants and provided a more equal distribution between ages;Socioeconomic status: low; high–the Brazilian Economic Classification System [15] was used to classify the economic profile of the families into two levels; low economic level if below or equal to C2, and high economic level if higher or equal to C1;Ethnicity: white or non-white;Type of cancer: solid (i.e., osteosarcomas, Wilms’ sarcoma, neuroblastomas, cerebral tumors, liver tumor, mediastinal tumor) or non-solid (i.e., leukemias and lymphomas);Current treatment time: ≤1 year or >1 year;Treatment phase: treatment or non-treatment phase–to be assigned to the non-treatment phase the participant was required not to be receiving any venous chemotherapy, radiotherapy, or surgery, and still be in the period within 5 years of cancer diagnosis;Type of treatment: chemotherapy, radiotherapy, surgery (yes or no);Physical therapy: yes or no; andType of assistance: public health system, or no public health system.

### 2.4. Dependent Variable

To evaluate the functioning of the participants, the Pediatric Evaluation of Disability Inventory-Computer Adaptive Test (PEDI-CAT) questionnaire was used [16]. This questionnaire is translated and validated into the Portuguese language [17].

The PEDI-CAT is composed of four domains: daily activities, mobility, social/cognitive, and responsibility [16]. This instrument aims to provide a detailed description of the individual’s performance and to document the individual changes and progress of the functional skills acquired after an intervention. The PEDI-CAT is not a performance-based “test,” but rather a large bank of 276 functional activities acquired during childhood and adolescence. Application of the questionnaire requires a computer with the instrument software installed and it can be self-administered or conducted by a professional through an interview with the parents to ensure understanding of the information for each item [16].

Four autonomous domains are displayed, which can be administered alone or with the other domains. Total points represent scaled scores which represent a way of looking at the difficulties the child has and progressing in skills over time. In the domains of daily activities, mobility, and social/cognitive, the score is based on an ordinal scale of four points with different levels of difficulty. The responsibility domain classifies items on a five-point scale, describing the sharing of responsibility between the caregiver and the child or adolescent in the completion of each item. For the four domains, the respondent is required to choose the option that best describes the child’s role for each item. If the respondent is not sure, there is an option to respond, “I do not know” [16].

In the current study, normative scores based on the chronological age of the child were used. Scores between 30 and 70 are considered adequate according to the age range of the child and/or adolescent: the scores were categorized into normal (score ≥ 30) and inferior performance (score < 30) [16].

### 2.5. Statistical Analysis

Descriptive analyses of the sample regarding the control variables are presented. Normality of the data was checked with Kolmogorov–Smirnov analysis and not confirmed. In addition, the mean, standard deviation, confidence interval, and standard error of measurement of the PEDI-CAT normative score of each domain were calculated.

A logistic regression analysis was used to verify the association of functioning with socioeconomic and demographic variables, type of cancer, type and stages of treatment, physical therapy, and type of health care factors. The univariate logistic regression analysis was performed for each functional domain of the PEDICAT, and all variables with a *p*-value inferior to 0.20 were eligible for multivariable analysis. For the final model of logistic regression, the variable was considered significant when it reached the level of significance below 0.05. Statistical Package for Social Sciences (SPSS, v. 23, 2018) was used for all analysis.

## 3. Results

Contact details were obtained of 102 potential participants. Of these, 69 were excluded because of age (over 18 y or under 2 y), inability to contact them (non-existent telephone number), difficulty traveling to the city for data collection, or refusal to participate. Finally, 33 children and/or adolescents agreed to participate in the study. The participants had a mean age of 9.23 years, 57.6% were female, 54.5% were younger than 8 years and had low socioeconomic status, 69.7% had non-solid tumors, and 63.6% were undergoing treatment.

The PEDI-CAT domains presented the following results: (1) Daily activities: 42.76 (12.54; CI 95% 38.31–47.21; SME 2.18); (2) Mobility: 32.93 (16.07; CI 95% 27.23–28.64; SME 2.79); (3) Social/cognitive: 40.06 (12.57; CI 95% 35.60–44.51; SME 2.18); and (4) Responsibility: 39.70 (10.16; CI 95% 36.09–43.30; SME 1.77). The mobility domain presented an average value very close to the lower limit considered suitable for age. Considering the values of confidence interval and standard error of the measure, the mobility domain presented worse functioning.

Table 1, Table 2, Table 3 and Table 4 present the frequency and percentages of the control variables of the participants. In addition, the tables show the results of the logistic regression analysis (odds ratio, 95% confidence interval, and *p*-value) regarding the number of children with inferior and normal performance in each of the PEDICAT domains according to each control variable used in the model. Still following the analysis according to the normative PEDI-CAT, multivariate logistic regression analysis showed that only the age control variable, in the mobility domain of the PEDI-CAT, presented a significant association. The participants older than 8 years presented worse mobility (OR = 22.000, 95% CI = 3.415–141.733, *p* = 0.0001). None of the other variables showed significant results.

## 4. Discussion

This study reports the impact of childhood and juvenile cancer on the functioning of children and adolescents diagnosed with cancer. To date, few studies have investigated the effects of this health condition on the functioning of children and adolescents with cancer, and almost all the manuscripts published on this topic focus on children with leukemia [18,19,20].

The present study showed that children and young people over 8 years of age present a chance of demonstrating mobility that is 22-times worse than children below 8 years of age. This result likely reflects the effects that the disease and treatment exert or exerted on these older children. It is possible these children did not receive any physiotherapeutic interventions during the period of hospitalization with a focus on maintaining the mobility of these individuals. Only 11 (33.3%) participants in the study received some type of physical therapy intervention during their treatment. In addition, since they only received respiratory care, it is probable the physiotherapy treatment (control variable) did not impact on any domain of the PEDI-CAT. Guidelines specific for children and adolescents with cancer are required with the aim of ensuring the child remains active throughout the therapeutic process. It is now known that in adult patients it is essential to practice physical exercises before, during, and after cancer treatment, with a frequency of 150 min per week of moderate intensity [21,22]. With active practice, adults with cancer can maintain their physical activity levels, preserving mobility during the treatment. Unfortunately, this recommendation does not yet exist for children and adolescent populations.

To date, studies mainly focus on children and adolescents with leukemia. In the present study, this group represents less than 70% of the types of cancer. Despite the low number of up-to-date studies that investigate functioning in children during the treatment or control phase of cancer [18,19,20], a systematic review and meta-analysis study of randomized controlled trials showed that the mobility of children with cancer is impaired during and after treatment and concluded that physical training can improve the mobility of these children [23]. Scientific evidence shows that survivors of childhood cancer tend to be more sedentary because they experience severe fatigue due to chemotherapy and radiotherapy treatments [24,25]. A meta-analysis brought together studies showing that both physical activity and physical fitness were significantly less frequent in childhood cancer survivors than in non-cancer controls [26].

If the impact of loss of physical activity is so prominent after the cancer is cured, it would be expected that during the treatment phase the effects of the treatment would be even worse. In the present study, only mobility showed a significant difference in the functioning profile of the children and adolescents with cancer, especially after the age of eight. These results are probably due to lack of activity, overprotection of parents, the course of the disease, and/or the cancer treatment or drugs which cause fatigue [27,28]. A recent study conducted a physical intervention program including twice-weekly cardiorespiratory and muscle strength training. The authors concluded that increased physical activity is longitudinally associated with less cancer-related fatigue; therefore, it is important to encourage physical activity in childhood cancer survivors [29].

Daily activities, social/cognition, and responsibility did not present differences between ages, or any other control variable reported in this study. The literature does not provide other studies to support the results of the present study. It is possible that the effect of the cancer treatment has less or no effect on these domains during the cancer treatment phase. As the literature is poor in studies regarding these variables, we recommend that these results be interpreted with caution. Future studies are encouraged which investigate these domains during treatment, in larger samples and in cancer survivor patients.

The current study presents some limitations, such as the small number of participants; however, all the adolescents and children diagnosed with cancer who were enrolled in the main foundation to support children’s cancer in the region that the team was able to contact during the period of accomplishment of the study participated. Future studies are encouraged, including more patients with cancer, across institutions from Brazil or around the world, using a cluster design or prospective design, to confirm or refute the results presented by this study.

## 5. Conclusions

The current study showed that children and adolescents older than 8 years, with different types of cancer demonstrated a higher chance of presenting lower mobility than expected for their normal peers at this age, and compared with children under 8 years old. This study may guide physical therapy interventions essential for the maintenance, development, preservation, and improvement in the mobility capacity and performance of children and adolescents with cancer.

## Figures and Tables

**Table 1 ijerph-19-09762-t001:** Frequency, number of children with inferior and normal performance in PEDICAT-Daily Activities, odds ratio, 95% confidence interval, and *p*-value.

Variables	Frequency (*n* = 33)		PEDICAT-DA Inferior Performance		PEDICAT-DA Normal Performance		OR	95% CI	*p*-Value
	N	%	N	%	N	%			
**Sex**									
Male	14	42.4	3	60.0	11	39.3	Ref		
Female	19	57.6	2	40.0	17	60.7	0.431	0.062–3.012	0.396
**Age**									
<8 years	18	54.5	1	20.0	17	60.7	Ref		
≥8 years	15	45.5	4	80.0	11	39.3	6.182	0.608–62.831	0.124
**Socioeconomic level**									
Low	18	54.5	3	60.0	15	53.6	Ref		
High	15	45.5	2	40.0	13	46.4	0.769	0.111–5.338	0.791
**Ethnicity**									
White	14	42.4	4	80.0	10	35.7	Ref		
Not white	19	57.6	1	20.0	18	64.3	0.139	0.014–1.418	0.096
**Type of Cancer**									
Solid	10	30.3	2	40.0	8	28.6	Ref		
Not solid	23	69.7	3	60.0	20	71.4	0.600	0.084–4.294	0.611
**Current treatment time**									
≤1 year	17	51.5	4	80.0	13	46.4	Ref		
>1 year	16	48.5	1	20.0	15	53.6	0.217	0.021–2.191	0.195
**Treatment Phase**									
Treatment	21	63.6	3	60.0	18	64.3	Ref		
Non-treatment	12	36.4	2	40.0	10	35.7	1.200	0.171–8.426	0.855
**Radiotherapy**									
No	28	84.8	5	100.0	23	82.1	Ref		
Yes	5	15.2	0	0.0	5	17.5	0.000	-	0.999
**Surgery**									
No	23	69.7	4	80.0	19	67.9	Ref		
Yes	10	30.3	1	20.0	9	32.1	0.528	0.051–5.427	0.591
**Chemotherapy**									
No	4	12.1	0	0.0	4	14.3	Ref		
Yes	29	87.9	5	100.0	24	85.7	-	-	0.999
**Physical Therapy**									
No	22	66.7	3	60.0	19	67.9	Ref		
Yes	11	33.3	2	40.0	9	32.1	1.407	0.199–9.963	0.732
**Type of Assistance**									
Public health system	29	87.9	3	60.0	26	92.9	Ref		
No public health system	4	12.1	2	40.0	2	7.1	8.667	0.873–86.062	0.065

Legends: PEDICAT-DA = Pediatric Evaluation of Disability Inventory Computer Adaptive Test-domain Daily Activities; *n* = number; OR = odds ratio; 95%CI = 95% Confidence Interval; *p*-value = level of statistical significance; Ref = reference category.

**Table 2 ijerph-19-09762-t002:** Frequency, number of children with inferior and normal performance in PEDICAT-Mobility, odds ratio, 95% confidence interval, and *p*-value.

Variables	Frequency (*n* = 33)		PEDICAT-M Inferior Performance		PEDICAT-M Normal Performance		OR	95% CI	*p*-Value
	N	%	N	%	N	%			
**Sex**									
Male	14	42.4	5	38.5	9	45.0	Ref		
Female	19	57.6	8	61.5	11	55.0	1.309	0.316–5.431	0.711
**Age**									
<8 years	18	54.5	2	15.4	16	80.0	Ref		
≥8 years	15	45.5	11	84.6	4	20.0	22.000	3.415–141.733	0.001 *
**Socioeconomic level**									
Low	18	54.5	9	69.2	9	45.0	Ref		
High	15	45.5	4	30.8	11	55.0	0.364	0.084–1.583	0.178
**Ethnicity**									
White	14	42.4	5	38.5	9	45.0	Ref		
Not white	19	57.6	8	61.5	11	55.0	1.309	0.316–5.431	0.711
**Type of Cancer**									
Solid	10	30.3	5	38.5	5	25.0	Ref		
Not solid	23	69.7	8	61.5	15	75.0	0.533	0.118–2.408	0.414
**Current treatment time**									
≤ 1 year	17	51.5	6	46.2	11	55.0	Ref		
> 1 year	16	48.5	7	53.8	9	45.0	1.426	0.351–5.793	0.620
**Treatment Phase**									
Treatment	21	63.6	9	69.2	12	60.0	Ref		
Non-treatment	12	36.4	4	30.8	8	40.0	0.667	0.152–2.926	0.591
**Radiotherapy**									
No	28	84.8	11	84.6	17	85.0	Ref		
Yes	5	15.2	2	15.4	3	15.0	1.030	0.148–7.193	0.976
**Surgery**									
No	23	69.7	9	69.2	14	70.0	Ref		
Yes	10	30.3	4	30.8	6	30.0	1.037	0.227–4.728	0.963
**Chemotherapy**									
No	4	12.1	2	15.4	2	10.0	Ref		
Yes	29	87.9	11	84.6	18	90.0	0.611	0.075–4.983	0.646
**Physical Therapy**									
No	22	66.7	8	61.5	14	70.0	Ref		
Yes	11	33.3	5	38.5	6	30.0	1.458	0.335–6.347	0.615
**Type of Assistance**									
Public health system	29	87.9	11	84.6	18	90.0	Ref		
No public health system	4	12.1	2	15.4	2	10.0	1.636	0.201–13.344	0.646

Legends: PEDICAT-M = Pediatric Evaluation of Disability Inventory Computer Adaptive Test-domain Mobility; *n* = number; OR = odds ratio; 95% CI = 95% Confidence Interval; *p*-value = level of statistical significance; Ref = reference category; * *p* < 0.05.

**Table 3 ijerph-19-09762-t003:** Frequency, number of children with inferior and normal performance in PEDICAT-Social Cognitive, odds ratio, 95% confidence interval, and *p*-value.

Variables	Frequency (*n* = 33)		PEDICAT-SC Inferior Performance		PEDICAT-SC Normal Performance		OR	95% CI	*p*-Value
	N	%	N	%	N	%			
**Sex**									
Male	14	42.4	3	37.5	11	44.0	Ref		
Female	19	57.6	5	62.5	14	56.0	1.310	0.255–6.715	0.746
**Age**									
<8 years	18	54.5	3	37.5	15	60.0	Ref		
≥8 years	15	45.5	5	62.5	10	40.0	2.500	0.485–12.886	0.273
**Socioeconomic level**									
Low	18	54.5	5	62.5	13	52.0	Ref		
High	15	45.5	3	37.5	12	48.0	0.650	0.127–3.325	0.605
**Ethnicity**									
White	14	42.4	3	37.5	11	44.0	Ref		
Not white	19	57.6	5	62.5	14	56.0	1.310	0.255–6.715	0.746
**Type of Cancer**									
Solid	10	30.3	3	37.5	7	28.0	Ref		
Not solid	23	69.7	5	62.5	18	72.0	0.648	0.121–3.466	0.612
**Current treatment time**									
≤1 year	17	51.5	4	50.0	13	52.0	Ref		
>1 year	16	48.5	4	50.0	12	48.0	1.083	0.220–5.326	0.922
**Treatment Phase**									
Treatment	21	63.6	7	87.5	14	56.0	Ref		
Non-treatment	12	36.4	1	12.5	11	44.0	0.182	0.019–1.706	0.136
**Radiotherapy**									
No	28	84.8	7	87.5	21	84.0	Ref		
Yes	5	15.2	1	12.5	4	16.0	0.750	0.071–7.883	0.811
**Surgery**									
No	23	69.7	6	75.0	17	68.0	Ref		
Yes	10	30.3	2	25.0	8	32.0	0.708	0.116–4.318	0.708
**Chemotherapy**									
No	4	12.1	1	12.5	3	12.0	Ref		
Yes	29	87.9	7	87.5	22	88.0	0.955	0.085–10.710	0.910
**Physical Therapy**									
No	22	66.7	4	50.0	18	72.0	Ref		
Yes	11	33.3	4	50.0	7	28.0	2.571	0.500–13.229	0.258
**Type of Assistance**									
Public health system	29	87.9	7	87.5	22	88.0	Ref		
No public health system	4	12.1	1	12.5	3	12	1.048	0.093–11.754	0.970

Legends: PEDICAT-SC = Pediatric Evaluation of Disability Inventory Computer Adaptive Test-domain Social Cognitive; *n* = number; OR = odds ratio; 95% CI = 95% Confidence Interval; *p*-value = level of statistical significance; Ref = reference category.

**Table 4 ijerph-19-09762-t004:** Frequency, number of children with inferior and normal performance in PEDICAT-Responsibility, odds ratio, 95% confidence interval, and *p*-value.

Variables	Frequency (*n* = 33)		PEDICAT-R Inferior Performance		PEDICAT-R Normal Performance		OR	95% CI	*p*-Value
	N	%	N	%	N	%			
**Sex**									
Male	14	42.4	4	57.1	10	38.5	Ref		
Female	19	57.6	3	42.9	16	61.5	0.469	0.086–2.547	0.380
**Age**									
<8 years	18	54.5	2	28.6	16	61.5	Ref		
≥8 years	15	45.5	5	71.4	10	38.5	4.000	0.648–24.693	0.136
**Socioeconomic level**									
Low	18	54.5	6	85.7	12	46.2	Ref		
High	15	45.5	1	14.3	14	53.8	0.143	0.015–1.360	0.090
**Ethnicity**									
White	14	42.4	3	42.9	11	42.3	Ref		
Not white	19	57.6	4	57.1	15	57.7	0.978	0.181–5.283	0.979
**Type of Cancer**									
Solid	10	30.3	3	42.9	7	26.9	Ref		
Not solid	23	69.7	4	57.1	19	73.1	0.491	0.087–2.770	0.421
**Current treatment time**									
≤1 year	17	51.5	4	57.1	13	50.0	Ref		
>1 year	16	48.5	3	42.9	13	50.0	0.750	0.139–4.035	0.738
**Treatment Phase**									
Treatment	21	63.6	5	71.4	16	61.5	Ref		
Non-treatment	12	36.4	2	28.6	10	38.5	0.640	0.104–3.951	0.631
**Radiotherapy**									
No	28	84.8	7	100.0	21	80.8	Ref		
Yes	5	15.2	0	0.0	5	19.2	0.000	-	0.999
**Surgery**									
No	23	69.7	5	71.4	18	69.2	Ref		
Yes	10	30.3	2	28.6	8	30.8	0.900	0.143–5.662	0.911
**Chemotherapy**									
No	4	12.1	1	14.3	3	11.5	Ref		
Yes	29	87.9	6	85.7	23	88.5	0.783	0.069–8.934	0.844
**Physical Therapy**									
No	22	66.7	4	57.1	18	69.2	Ref		
Yes	11	33.3	3	42.9	8	30.8	1.687	0.304–9.358	0.549
**Type of Assistance**									
Public health system	29	87.9	5	71.4	24	92.3	Ref		
No public health system	4	12.1	2	28.6	2	7.7	4.800	0.540–42.632	0.159

Legends: PEDICAT-R = Pediatric Evaluation of Disability Inventory Computer Adaptive Test-domain Responsibility; *n* = number; OR = odds ratio; 95% CI = 95% Confidence Interval; *p*-value = level of statistical significance; Ref = reference category.

## Data Availability

Data available on request due to ethical permissions. The data presented in this study are available on request from the corresponding author.

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
