# Peer review of "Functioning of Children and Adolescents with Cancer"

_ijerph, 2022, doi:10.3390/ijerph19159762_

Round 1
Reviewer 1 Report
- The clarity of the . Discussion presentation needs to be improved
- Discussion needs more details so that the reader can understand the impact of these information.
- The authors stated the references which needs to reorganized.
Author Response
Dear reviewer 1:
Thanks for your kind review and following considerations that contributed to the new version of this manuscript. Your comments are highlighted in blue highlights, together with the recommendations of the reviewer 3.
- The Introduction and discussion were reviewed.
- The references were reorganized according to the norms of the journal.
- At last, the manuscript went through English review with a native English reviewer

Reviewer 2 Report
The aim of this study was to evaluate the functioning of children and adolescents diagnosed with cancer and verify the association of socioeconomic, demographics, type of cancer, type and stages of treatment, physical therapy and type of health care factors with functioning.
This research, although it does not provide very relevant results, is methodologically well thought out and very well written, easy to read.
This study did not have any limitations?
The table footers are not all the same. Unify the format.
Congratulations to the authors.
Author Response
Dear reviewer 2:
Thanks for your kind review and following considerations that contributed to the new version of this manuscript. Your suggestions follow in yellow highlight
- The limitations were added to the end of the discussion.
- And the headings and footnotes of the tables were unified.

Reviewer 3 Report
Dear Authors,
After thorough reading of your manuscript, I have the following recommendations:
1. Introduction is too long, it is not focused on the manuscript. It should reflect background information of the study, not of cancer. 4 paragraphs should be enough
2. Abstract is adequate, but can be improved.
3. Material and methods section is well written.
4. Results show only a statistical difference in terms of mobility. Do you have a possible explanation why other areas aren't influenced? Even if statistical significance is not achieve I believe it may be beneficial to adress these in the discussions section, otherwise the discussion section only focusses on mobility
5. Discussions sections I believe can be improved.
6. Limitations of the study are adressed. Do you believe that a prospective study would add more data? And if so, would you be able to perform a cluster study design across multiple institutions from Brazil/International as the next step?
English language: Further polishing is required. Some phrases are too long, use short, concise sentences instead.
Kind regards
Author Response
Dear reviewer 3:
Thanks for your kind review and following considerations that contributed to the new version of this manuscript. Your considerations follow in blue highlights, together with the recommendations from the reviewer 1.
- The abstract was reviewed.
- The introduction was reduced according to your suggestions.
- The discussion was rewritten and a paragraph regarding the absence of changes in the other domains of the PEDI-CAT was added.
- In the limitations, future studies were proposed.
- At last, the manuscript went through English review with a native English reviewer

Round 2
Reviewer 3 Report
Dear Authors,
I believe that this version of the manuscript solves the issues I raised.
Also, at line 47, "the" should be removed.
Kind regards.